# Checks and Balances: Enforcing Constitutional Constraints

**Randall G. Holcombe** 

Department of Economics, Florida State University, Tallahassee, FL 32306, USA; holcombe@fsu.edu

**Abstract:** Constitutional political economy has focused heavily on designing constitutional rules sufficient to constrain governmental power. More attention has been devoted to designing rules that are effective constraints than on the institutions that would be required to enforce them. One problem is that rules are interpreted and enforced by the political elite, who tend to interpret and enforce them in ways that favor their interests over those of the masses. Democratic oversight is ineffective because voters realize they have no influence over public policy, and are therefore rationally ignorant. A system of checks and balances within government is necessary for enforcing constitutional constraints because it divides power among elites with competing interests and enables one group of elites to check the power of others. Checks and balances within governmental institutions are necessary to constrain the government from abusing its power.

**Keywords:** constitutional constraints; checks and balances; political elite; democratic oversight

## 1. Introduction

Buchanan (1975, pp. 2–3, *italics in original*) discusses the distinguishing feature of constitutional economics. "In ordinary or orthodox economics, no matter how simple or how complex, analysis is concentrated on choices made *within* constraints that are, themselves, imposed exogenously to the person or persons charged with making the choice. Constitutional economics directs analytical attention to the *choice among constraints*". North (1991, p. 97) says "Institutions are the humanly devised constraints that structure political, economic, and social interaction". Constitutional economics, following Buchanan and North, studies the choice among institutions—the choice among humanly devised constraints. The literature in constitutional economics has focused heavily on the design of effective rules to prevent the abuse of government power, and to facilitate government production that benefits the general population rather than concentrated special interests. It has focused less on the design of institutions that are able to effectively enforce those rules. This paper explains why checks and balances are essential as a constitutional enforcement mechanism.

## 2. Constitutional Rules

Much of the work in constitutional economics focuses on the process by which citizens choose constitutional rules, with a heavy emphasis on consensus among those who are governed by them. The literature draws a parallel between market exchange, in which all parties to transactions agree and thereby signal that they benefit from their exchanges, and political exchange, in which people cooperate to collectively produce goods and services that they would not be able to produce by themselves or through bilateral exchange. Maintaining that parallel construction, the political decision rule that signifies everyone agrees is unanimity. Just as all parties to market transactions voluntarily agree to them, unanimous agreement is the signal that all parties to political exchange are in agreement.

The constitutional framework pioneered by Buchanan and Tullock (1962) and Buchanan (1975) explains that for governmental activity to benefit everybody, agreement is required on the rules under which people interact, but not generally on every action government takes. For example, people might all agree that they benefit from paying taxes to finance roads, and might agree to delegate the decision of where to build those roads to some governmental authority. In some cases, citizens might be made worse off by the building (and financing) of some specific roads, but they would all agree they are better off with the government-financed roads than they would be without them. In keeping with the analogy to market exchange, optimal rules are rules to which everybody would agree.

Most of the literature in constitutional economics has dealt with the process by which constitutional rules are designed, and the types of rules that would result from those processes. There are many interesting and unresolved issues in this literature, which this paper bypasses. Assuming that a desirable set of constitutional rules has been put into place, how can those rules be enforced? The question of enforcement starts with interpretation. In a complex world, words can be interpreted in different ways. An easy way to see that is to note that slightly more than half of United States Supreme Court cases have been decided unanimously, which means that in nearly half of those cases, legal experts—the Justices themselves—were in disagreement about how the law applies to specific cases.

Schweizer (2013) says that laws are written to be deliberately ambiguous, for several reasons. One reason is that ambiguous laws can be selectively enforced. Those who exercise political power can use ambiguous laws to go after adversaries but to give allies a pass. Another is that those who write ambiguous laws become experts on their intent and interpretation, allowing them to sell their services to those who might be subject to enforcement.

Ambiguities in interpretation point to a second issue: Selective enforcement. Selective enforcement allows laws to be enforced for the benefit of the enforcers and to the detriment of others who are in an adversarial position to the enforcers, or are not being as cooperative as the enforcers would prefer.[1] Factors like these have not been addressed in the constitutional political economy literature, which has for the most part assumed that rules are unambiguous and objectively applied.[2]

To the extent that enforcement issues have been considered, the primary mechanism that appears in the public choice literature is democratic oversight of government actions. While voting models tend to point toward public policy being determined by the preferences of the electorate—in particular, models like the median voter model explained by Downs (1957)—the conclusion of much of the public choice literature is that democratic oversight is likely to be ineffective for a number of reasons. Rent-seeking (Tullock 1967; Krueger 1974), regulatory capture (Stigler 1971), and the undue influence of special interests (Olson 1965) all weigh against the idea that democratic oversight is an effective check on the abuse of government power. Meanwhile, checks and balances within government institutions have been underappreciated in the literature as an enforcement mechanism. To see why democratic oversight is likely to be ineffective, and why checks and balances within government are essential, the first step is to undertake a critical examination of the process by which public policy is made and carried out in democratic governments.

---

[1]   Allison (2013), CEO of BB&T bank during the 2008 financial crisis, recounts banking regulators pressuring him into participating in the government's bailout program, even though he said his bank was financially sound and did not need a bailout. They told him that new regulations were being written, and while they did not know what those regulations would be, if he did not join the program, his bank would be in danger of being out of compliance. Allison took that as a threat that unless he cooperated, regulations would be written and enforced to target his bank.

[2]   There is a literature on corruption that considers these issues. Aidt (2016) notes that there are commonalities between this literature and the public choice literature on rent-seeking that have been left relatively undeveloped.

### 3. The Public Policy Process

Democratic governments are, by necessity, representative democracies. Governments that rule over millions of people, or even thousands, cannot be governed by direct democracy because transaction costs are too high to allow that many people to negotiate public policy. Even if referendums on public policy were offered to voters, the choices given to voters would be determined by an elite few who have been delegated to exercise government power. Sometimes the political elite get their power through democratic elections and sometimes through political appointment. Applying the Coase (1960) theorem, a small group of people face low transaction costs and can bargain to design public policy. They are the political elite. Most people face high transaction costs and are unable to participate in the policy-making process. They are the masses.

The US House of Representatives, for example, has 435 members, a small enough group that they all know each other and are able to engage in logrolling to produce policies that they view as beneficial. Most readers of this paper will find themselves in the high transaction cost group, and will be unable to participate in the political bargaining process that produces legislation. Public policy is created by a small group of people, and the resulting policies apply to everyone.

The public choice literature offers well-established explanations for how the political process can be used to benefit some while imposing costs on others. Rent-seeking, following Tullock (1967) and Krueger (1974), describes how some are able have resources transferred to them from others using the political process. Stigler (1971) has explained how some are able to capture the regulatory process so that regulations nominally designed to constrain their actions benefit those who are being regulated at the expense of others. Olson (1965, 1982) has explained how concentrated and well-organized interests are able to use the political process for their benefit, at the expense of larger groups that are less well-organized. All of these theories explain how some people are able to use the political process to benefit themselves at the expense of others.

While the public choice literature offers many explanations for how some can use the political process for their own benefit, it has not followed up by explaining that there is a relatively small and stable group of individuals who are able to command rents, capture regulatory agencies, and organize influential interest groups. One reason is that the public choice literature tends to take an individualistic approach to analyzing political processes, which limits its vision in identifying the specific group of people who benefit from political processes. Buchanan and Tullock (1962, p. 12) "reject any theory or conception of the collectivity which embodies the exploitation of a ruled by a ruling class. This includes the Marxist vision, which incorporates the polity as one means through which the economically dominant group imposes its will on the downtrodden". When one analyzes the political process, though, public policy is made by a few—a ruling class—and imposed on everyone. Some people are consistently in the group that benefits, and most people are consistently in the group that bears the costs.

Elite theory, developed mainly in sociology and political science, paints a different picture, describing ruled and ruling classes along the lines that Buchanan and Tullock dismiss. Mills (1956, p. 3) says "The powers of ordinary men are circumscribed by the everyday worlds in which they live . . . But not all men are in this sense ordinary. As the means of information and power are centralized, some men come to occupy positions in American society from which they can look down upon, so to speak, and by their decisions mightily affect, the everyday world of ordinary men and women". Those are the power elite—the few who make the public policies to which everyone else is required to conform. Elite theory explains who designs and controls public policy, but it does not explain how they are able to exercise this control. Public choice theory explains how some are able to use the system for their benefit at the expense of others, but it does not identify who those some are. Holcombe (2018) explains that when these two strands of academic literature are combined, they show that those who obtain the rents, capture regulatory agencies, and form powerful interest groups are a relatively stable group—the elite.

Elite control of the public policy process has not escaped observation by economists. Stiglitz (2012, pp. 39–40) says "We have a political system that gives inordinate power to those at the top, and they have used that power not only to limit the extent of redistribution but also to shape the rules of the game in their favor". Stiglitz (2012, p. 59) goes on to say, "It's one thing to win a 'fair' game. It's quite another to be able to write the rules of the game—and to write them in ways that enhance one's prospects of winning. And it's even worse when you can choose your own referees. It doesn't have to be this way, but powerful interests ensure that it is". Along these same lines, Acemoglu and Robinson (2008) develop a model in which democratic political institutions evolve to favor the elite.

Nor has elite control of the public policy process escaped observation by the general public. After the financial crisis that began in 2008, the Occupy Wall Street movement protested the government policies that bailed out the Wall Street financial firms that took losses on their mortgage-backed securities but did nothing to help people who found themselves under water on their mortgages and were unable to pay them because they had lost their jobs due to the recession. They were complaining about policies that they said favored the 1 percent over the 99 percent. In academic jargon, the 1 percent are the elite; the 99 percent are the masses.

To a certain extent, the actions of the 99 percent produced the very policies they were protesting. The twenty-first century view of the role of democratic government is to carry out the will of its citizens as revealed through a democratic political process.[3] When a crisis appeared in 2008, the 99 percent demanded that the government do something to mitigate the crisis. Looking at how the political process actually works, government is run by the 1 percent—the elite. Thus, the 99 percent were demanding that the 1 percent be given more power to take action in response to the financial crisis, which the 1 percent did. Understanding how the process works, one should not be surprised that when the 1 percent took action, the action they took furthered their own interests. The elite make public policy, so one should expect that when they find themselves relatively unconstrained, public policy works to the advantage of the elite.

This is, Brennan and Buchanan (1985) explain, the reason for rules. They explain that a constitutional framework provides the foundation for individual interaction. Those constitutional rules should be designed to create a framework that channels individual actions away from predatory zero-sum and negative-sum action toward action that is positive-sum and mutually advantageous. Brennan and Buchanan (1985, p. 5) reference Hobbes to say that we benefit from a set of rules that govern people's interactions with each other because " . . . without them we would surely fight. We would fight because the object of desire for one individual would be claimed by another. Rules define the private spaces within which each of us can carry out our own activities".

What Brennan and Buchanan do not say is that when some people write and interpret the rules, one would expect them to write rules that favor themselves, and interpret any rules in ways that favor themselves. Buchanan and Congleton (1998) conclude that rules that are relatively permanent and that apply generally to everyone will receive widespread approval from the masses. What this line of reasoning does not take into account is that the elite write, interpret, and enforce the rules. Writing the rules is the first step in the process. Even in this first step, those who write the rules must be constrained to write them in a way that benefits everyone, not just the elite who write them. Then, the elite must be constrained to interpret the rules in an even-handed manner that does not favor themselves over the masses. And then, rules must not be selectively enforced so that enforcement favors those who have political power over those who do not. The analysis that follows takes the first

---

[3] This contrasts with the vision of democracy held by the American Founders, who designed a government with constitutionally limited enumerated powers, and designed constraints to try to prevent government from actions not specifically permitted by the Constitution. In this view, democracy is a method of selecting who exercises the power of government, but not a method for determining what those powers are. The more modern view, Holcombe (2002) explains, envisions democratic governments as furthering the public interest, where the public interest is revealed through a democratic political process.

step as given, and examines how rules can be interpreted and enforced in a way that does not favor the elites (who interpret and enforce the rules) over the masses.

One might ask whether this framework, which depicts the elite as controlling the political process for their benefit, overstates their influence over public policy. There is an extensive literature written by authors ranging from the political right to the political left who make just this observation. Nader (2014) argues that objections to the coalition of economic and political elites are so widely recognized from one end of the political spectrum to the other that they form an unstoppable coalition that will put an end to the cronyism and corporatism that everyone observes.[4] Stockman (2013, p. 52) says, "Trying to improve capitalism, modern economic policy has thus fatally overloaded the state with missions and mandates far beyond its capacity to fulfill. The result is crony capitalism—a freakish deformation that fatally corrupts free markets and democracy". Bartels (2008) refers to this situation as the new gilded age. Gilens (2012) cites growing inequality in political power that is creating an increasingly divided society.

The idea that the political process is run by the elite for their benefit is well-supported in the academic literature. Even if that literature overstates the power of the elite to benefit themselves at the expense of the masses, there is still good reason to design an institutional framework in which the ability of some to use it to benefit themselves at the expense of others is limited. How can constitutional constraints on government power be enforced, when those who enforce them have an incentive to use their enforcement powers to benefit themselves?

## 4. Democracy

One mechanism for enabling the 99 percent to exercise oversight over the 1 percent is democracy. Democratic elections allow citizens to select who exercises the power of government, and create a process whereby citizens can peacefully replace those with political power if they are unsatisfied with their performance. Downs (1957) develops a model in which competition for elected office results in the election of candidates who run on the platform most preferred by the median voter. Holcombe (1989) explains that this model has often been used to conclude that governments do what the median voter most prefers.

Beyond the median voter model, voting models in public choice generally conclude that the collective choice of a group is determined by aggregating the votes of the individuals in the group. The implication is that the policy outcomes implemented by government are those chosen by the voters. Even in models showing perverse outcomes, such as McKelvey's (1976) demonstration that that, in general, there is no stable equilibrium in majority rule voting, the conclusion, Riker (1980) explains, is that political processes are unstable. This potential instability has been widely recognized since the beginnings of the subdiscipline of public choice, and Arrow (1963) begins his well-known book with an example of a cyclical majority. Yet Tullock (1982) observes that political outcomes appear to be much more stable than economic outcomes, so if economists can argue that equilibrium models are descriptive of market outcomes, surely the concept of equilibrium outcomes applies more to government than to markets.

One explanation for the apparent stability of political outcomes, given in the previous section, is that voters do not choose those outcomes. Public policy is chosen by an elite few—the 1 percent—not the electorate. The elite are a small group who are able to bargain with each other to produce public policies that are most favorable to themselves because, following Coase (1960), transaction costs are low within that elite group. When transaction costs are low, political exchange produces a stable outcome for the same reason market exchange produces a stable outcome. Voters do not decide public policy outcomes. The elite negotiate among themselves to produce outcomes most valuable to them.

---

[4]　Nader may be overly optimistic on the success of this coalition, because while there is widespread agreement that the elite control the political process, there is not agreement on the remedy. Those on the left tend to favor more government control, while those on the right see the problem as being caused by government and argue for less government interference in economic affairs.

Just as with externalities in markets, the masses are in a high transaction cost group and are unable to bargain to prevent costs from being imposed on them.

Public choice theory offers many reasons to question whether voters really do exercise any effective control over the politicians they elect. One reason, given by Downs himself, is that it would be so rare for any election outcome (except with a very small number of voters) to be determined by a single vote that individual voters have no incentive to become informed. They are rationally ignorant because they realize their one vote will have no effect. Uninformed voters are not in a position to exercise control over those who they elect. Elections have symbolic value to elected officials by making it appear that what they do has been chosen by the electorate, Edelman (1964) observes, but all this does is give those who hold political power even more discretion to act to further their own interests rather than in the interests of those who elected them. Elected officials can claim that they are carrying out the will of the people, as revealed through the democratic political process.

Even those who are very interested in following politics will be unable to effectively monitor those who they elect. For one thing, being a politician is a full-time job, so anyone really interested in monitoring elected representatives would have to devote full time to it. Because there are many elected representatives, there would not be enough time to monitor all of them. Furthermore, the people who participate in political decision-making specialize in it, and have an understanding beyond what outsiders could hope to achieve. Just as citizens would not expect to have as much medical knowledge as a doctor, or as much knowledge about the operation of an automobile as an auto mechanic, it is unrealistic to expect citizens to have sufficient knowledge to monitor those who specialize in politics.

While there is a market and individual choice for those who are looking for medical services or auto repair services, there is no similar market for politicians. Those who exercise political power claim a monopoly over it, and while they can be challenged in elections, challengers are only making claims about what they would do if elected, so their claims cannot be verified as with incumbents who are actually practicing politics. Voters cannot observe their actual performance until after they have been elected, and few people think that campaign promises are as credible as, for example, automobile advertisements. Voters do not have reliable information, and even when such information is available, they have an incentive to remain rationally ignorant.

Because the general public has little incentive to organize to further their political interests, Olson (1965) explains that concentrated interests are able to effectively organize to provide political benefits to themselves by imposing costs on the masses. Public policy tends to favor special interests—the 1 percent—rather than working in the interest of the general public—the 99 percent. A well-established body of public choice literature helps explain why democratic oversight is likely to be an ineffective mechanism for enforcing constitutional constraints on government actions.

Public choice voting models nearly always assume that voters vote instrumentally; that is, they vote as if their votes can affect an election's outcome. But Downs' (1957) rational ignorance hypothesis rests on the conclusion that voters know their individual votes will not affect an election outcome, so they have no incentive to become informed or to vote instrumentally. Citizens do vote in large numbers, and Brennan and Lomasky (1993) conclude that because they know their individual votes will not affect election outcomes, they tend to vote expressively rather than instrumentally. They vote for outcomes that make them feel good rather than those that actually are in their individual interests, which Tullock (1971) notes can result in outcomes antithetical to their interests. Caplan (2007) goes one step further to argue that voters cast votes that are rationally irrational. They have no incentive to rethink any irrational public policy beliefs they hold, because their individual opinions will have no effect on public policy.

Wittman (1989, 1995) challenges the assessment given above, arguing that there are many mechanisms that direct public policy to follow the preferences of the voters. Political advertising and party brand name identification help provide voters with information, and voters can join special interest groups like the National Education Association and the National Rifle Association to have their collective preferences represented. Wittman explicitly notes that there are counter-arguments to all his

point. Indeed, he recognizes that he is presenting counter-arguments to generally-accepted models in public choice.

The point in mentioning Wittman's analysis is to note that while the arguments presented above about the influence of concentrated interests, rationally ignorant voters, and expressive voting are well-accepted in public choice, there are arguments going the other way. Despite any counter-arguments, the public choice literature provides many reasons for thinking that democratic oversight will not be an effective way to interpret and enforce constitutional rules. The essential point is that constitutional rules are designed, interpreted, and enforced by an elite few—the 1 percent—and the masses—the 99 percent—have essentially no say over them. One cannot expect the powerless 99 percent to police the activities of the elite. Public choice theory gives ample reasons why democratic oversight will be an ineffective constraint.

## 5. Checks and Balances

One of the celebrated innovations embodied in the Constitution of the United States was a set of checks and balances that enabled one branch of government to check and balance the power of the others. James Madison, in *Federalist No. 51* (1788), discusses the role of checks and balances, and explicitly recognizes that the reason they are required is to counter the potential for elite control of government to oppress the masses, saying "Ambition must be made to counter ambition . . . It may be a reflection of human nature, that such devices should be necessary to control the abuses of government". Constitutional rules will not be constraining unless those who interpret and enforce them are also constrained, and if an elite few interpret and enforce the rules, any checks on the power of those elites must come from other elites. The masses do not have the power to constrain elites, either through democratic oversight or by other means.

Persson et al. (1997) note that a system of checks and balances requires a separation of powers within the structure of government. But separation of powers is not the same as checks and balances, and by itself can lead to outcomes worse for the masses. Brennan and Hamlin (1994) show that if powers are merely separated, that can give those with some powers the ability to act unilaterally to the detriment of all others. Separation of powers can create a common pool problem, where some can use their (separate) powers for their benefit, imposing costs on others—others in different branches of government, and others in the broader citizenry.

Checks and balances mean that along with a separation of powers, one branch of government cannot act unilaterally without the agreement of another. But even this is not sufficient. The different branches of government should be designed so that they have conflicting interests, but must reach an agreement to take collective action. If their interests are all the same, they will act together to accomplish their common ends rather than checking and balancing each other. With conflicting interests, the interests of one branch can then check the interests of another. Persson et al. (1997) provide an example of an executive branch that can propose a total budget and a legislative branch that determines the components of the budget. By constraining the total size of the budget, the executive branch can check the legislative branch's ability to enlarge its individual components, and by controlling its components, the legislative branch can check the executive branch's appetite for expenditures.

La Porta et al. (2004) note that an independent judiciary and constitutional review work as a judicial check on the abuse of power by other branches of government. Different elements of government can act as "veto players" who can prevent other elements from unilateral action. Keefer (2002) looks at the effects of veto players on the ability of a central bank to undertake independent monetary policy, and Beck et al. (2002) provide data on veto players in government. This literature on veto players points to a productive way to view checks and balances, but veto players are not necessarily the product of constitutional design.

For example, when a parliament contains members of many parties, a coalition of parties will be required to take action, making the coalition members veto players. But, if a single party gains a

majority of seats, it eliminates other parties as veto players, even though the constitutional design remains unchanged. Ideally, a system of checks and balances will be built into the constitutional structure rather than be the result of political factionalization. Durable checks and balances are a part of the institutional structure within which government operates.

The system of checks and balances works on the principle that the individual branches of government guard their powers from being usurped by other branches. The key feature here is that some elites check and balance the power of others. If the masses have a minimal ability to check the power of government, the only way checks and balances can be effective is if institutions separate elites into groups with competing interests that have veto power over the actions of each other.

The United States Constitution embodies this idea by establishing executive, legislative, and judicial branches of government that can check and balance each other. As Madison said, ambition must be made to counter ambition. Actions taken by one branch require the cooperation of the others, and as originally conceived, the House of Representatives and Senate were designed as a part of this system. House members were elected by citizens and Senators were chosen by their state governments, with the idea that for legislation to pass, it had to be approved by the representatives of the people in the House and the representatives of the state governments in the Senate, as described by Zywicki (1997). This check was nullified in 1913 by the 17th Amendment to the Constitution, which required direct election of Senators. While Tarabar and Hall (2015) note that it appeared to have little immediate effect, the fact that its supporters pursued the difficult process of amending the Constitution indicates that they expected this weakening of checks and balances to make a difference. By eliminating this element of competing interests, the elite were able to remove one point of conflict that could act as a check on their abuse of power.

The US Constitution took some inspiration from British government, where the power of the crown was checked and balanced by the House of Lords and the House of Commons. Courts also had a place in checking the powers of the crown and Parliament. A comparison of British and American checks and balances shows that the functional division of power may play a minor role when compared to designing a system in which one group of elites has the ability to check and balance the power of others. Congleton (2012) describes the evolution of liberal political institutions in Europe over the last several hundred years, The key point is that over time institutions evolved to create political systems in which there was a division of power, and in which no branch of government could act independently without the cooperation of others.

With regard to contemporary American politics, Mann and Ornstein (2012) argue that constitutional checks and balances have eroded substantially since the nation's founding, and especially beginning in the twentieth century with increasing authority moving into the executive branch of government.[5] More generally, Acemoglu et al. (2013) argue that checks and balances have been eroded because the economic elite are better organized and are better able to influence politicians absent those checks and balances internal to the operation of government.

## 6. Other Checks and Balances

The checks and balances discussed above apply to a single government that is designed so that its various branches can check and balance each other. Madison, in *Federalist No. 51* (1788) says "In the compound republic of America, the power surrendered by the people is first divided between two distinct governments, and then the portion allotted to each subdivided among distinct and separate departments. Hence a double security arises to the rights of the people". An additional check is a federal system of government in which the state governments check the power of the federal government and the federal government checks the power of the states. This check was

---

[5]　Mann and Ornstein (2012) attribute much of the breakdown of checks and balances to the increase in partisan extremism, especially with reference to the Republican party, but that is beside the point for present purposes.

clearly embodied in the original Constitution of the United States by specifying that Senators would be chosen by their state governments, so for any legislation to be approved by both Houses to become law, it would have to meet with the approval of the representatives of the state governments, as Ostrom (1971) explains. This check was eliminated by the passage of the 17th Amendment in 1913, which specified that Senators be elected by a direct vote of the citizens.[6]

Intergovernmental competition can also provide a check on the powers of competing governments, Tiebout (1956) explains, pushing governments to respond to the demands of their citizens, yielding another benefit of a federal system. Decentralized political systems allow more local control, another possible check on the power of government. Local elites still control local government, but there is less distance, socially as well as geographically, between local elites and the general public. Local control, intergovernmental competition, and the ability of one level of government to check another are mechanisms that a federal system provides to check and balance the power of the elites who hold political power.

A free press is another mechanism that checks the power of government, which was recognized by the American Founders and embodied in the First Amendment to the Constitution. Coyne and Leeson (2009) note the impact of a free press on institutions, both reinforcing institutions that provide general benefits and undermining institutions that allow elites to abuse their power. This is a check that rests outside of government, but that can be enabled or inhibited by government control of the media.

## 7. Conclusions

The important role constitutional constraints on government play in protecting the rights and well-being of citizens has been well-recognized for centuries. Hume [1777] (1987, Essay VI) says "Political writers have established it as a maxim, that, in contriving any system of government, and fixing the several checks and controuls of the constitution, every man ought to be supposed a knave, and to have no other end, in all his actions, than private interest . . . Without this, they say, we shall in vain boast of the advantages of any constitution, and shall find, in the end, that we have no security for our liberties or possessions, except the good-will of our rulers; that is, we shall have no security at all". Constitutional political economy has focused heavily on designing rules that give those who hold political power the incentive to act in the public interest, but given those rules, they can only be effective if they are effectively enforced.

Checks and balances are a requirement for enforcement, because an elite few write, interpret, and enforce the rules. The 99 percent cannot regulate a process that is run by the 1 percent. The role of checks and balances is to have subsets of the 1 percent check and balance the power of others in that elite group.

The twenty-first century ideology of Progressive Democracy has weakened the constitutional constraints on government, because it justifies government policies that benefit some at the expense of others, and because it legitimizes the actions of a democratic government by depicting those actions as carrying out the will of the citizens, as revealed through a democratic political process. The ideology of twenty-first century Progressive Democracy encourages the 99 percent to demand the government take action to address a variety of issues, rarely recognizing that the 99 percent are transferring additional power to the 1 percent who write, interpret, and enforce the rules. Then, the 99 percent are surprised when the 1 percent uses their additional power for their own benefit, and in response the 99 percent again demand that the government should do something, which transfers even more power to the 1 percent.

---

[6]     The Articles of Confederation, the original US constitution, designed a government with only one legislative body whose members were chosen by the state governments, providing even more of a check on the power of the federal government.

Democracy is an ineffective constraint on abuse of government power, because it is based on the illusion that the 99 percent can exercise control over the 1 percent—the elite who actually design and enforce public policy. Public choice theory explains that the 99 percent are rationally ignorant (Downs 1957), that concentrated interests are able to use the political system for their benefit at the expense of the masses (Olson 1965), that government regulation works for the benefit of those who are regulated (Stigler 1971), and that the elite are able to design institutions to transfer power to themselves and away from the masses (Acemoglu and Robinson 2008). One cannot expect the powerless to control the powerful, even if the powerless well-outnumber the powerful.

The elite control government, so the most effective way to constrain government and enforce constitutional rules is to design institutions that give some elites the power to check the power of others. Checks and balances work to enforce constitutional rules through a separation of power, so that no single elite group can act without the cooperation of others. Separation of powers is not enough. Institutions must be designed so that elites have conflicting interests that give them the incentive to protect their own power by checking abuse of power by other elites. Institutions must be designed so that any abuse of power by one set of elites can be countered by the power of another set, and those different sets of elites must have the incentive to counter, to protect their own power.

The arguments presented here rest on a foundation that depicts government as ruled by an elite few, with the masses having essentially no power to design, interpret, and enforce the rules that constrain government. A more democratic vision of government depicts government as controlled through democratic processes that begin with elections in which the masses vote to elect the elite few who are able to exercise the power of government. Thus, in evaluating these arguments, one should consider which vision of government appears to be more descriptive of actual political institutions.

The constitutional political economy literature, tracing its origins back to Buchanan and Tullock (1962) and Buchanan (1975), depicts a set of constitutional constraints designed by a process that requires agreement among those who are subject to those constraints. Yeager (1985, 2001) argues that these models of hypothetical collective agreement to some set of rules have the pernicious effect of making government actions that are taken by an elite few and that are based on force appear as if they have been somehow agreed to by citizens. Recognizing that all government action is backed by the threat of force for noncompliance, and recognizing that an elite few are in a position where they can exercise government power, there is good reason for constitutional political economy, as a subdiscipline, to focus more attention on the types of institutions that can objectively interpret and enforce constitutional rules. It is not enough to have good rules if those rules are not objectively interpreted and enforced. If the elite really do control the state, there is good reason to think that democratic oversight will be ineffective and that checks and balances are necessary to constrain those who hold the power of government from abusing that power.

**Funding:** This research received no external funding.

**Conflicts of Interest:** The author declares no conflict of interest.

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
