# Peer review of "Checks and Balances: Enforcing Constitutional Constraints"

_economies, doi:10.3390/economies6040057_

Round 1
Reviewer 1 Report
Report is attached
Author Response
This review is more negative than the other two I received on the paper. I have tried to address the reviewer's comments, but sometimes found it difficult because the reviewer's comments seem to be addressed in the previous draft. I may have made fewer edits to the paper than this reviewer wanted, partly for that reason and partly because the other reviewers checked "Yes" in all the places this reviewer checked "Must Be Improved."
I did rewrite the abstract (reviewer point 1), and I appreciate the comments on the literature on veto players (reviewer point 5). I added the references the reviewer listed along with a discussion of the relevance of veto players to checks and balances.
With regard to point 4, it may be possible to undertake an empirical analysis, but that would be beyond the scope of the current paper, especially considering the editor has given me ten days to resubmit the paper.
Point 3 says the paper should be clear about its contribution. I did add some edits to say that the two contributions of the paper, as I see it, are the emphasis on the importance of checks and balances as institutional requirements to interpret and enforce constitutional rules, and relating this idea to the literature on constitutional economics, which rarely recognizes the essential role of checks and balances. The paper goes into an extended discussion to explain that this has been neglected in the constitutional political economy literature because it has not recognized the role of a limited group of elites who control the design of public policy.
Point 2 asks four questions that are related to the paper, but only peripherally related to its main ideas. It would be too much of a digression to answer them more fully in the paper, but consider each one:
a. Rules can provide constraints but not be implemented if there is no enforcement. Consider a Stop sign on a road. The rule is a constraint that requires cars to stop when they get to one, but if the rule is not enforced, cars will roll though them. Speed limits on roads are another example. They are designed to constrain the speed of a car, but if they are not enforced, cars will violate the rule. Do these examples clarify the question?
b. The question is, "Is implantation related to specific development, political or social outcomes or all of them?" I assume the reviewer means implementation, not implantation, but implementation is related to rules, not outcomes.The constitutional political economy literature is oriented toward optimal procedures, not outcomes, with the idea that the optimal procedure will result in the optimal outcome.
c. Checks and balances are not a part of democracy, as democracy is normally defined. Democracy is a form of government that is accountable to its citizens through elections and other institutional mechanisms. Checks and balances exist within the government that is being held accountable. Checks and balances are internal controls, whereas democracy is an external control. The reviewer says "checks and balances are part of democracy," but looking at it this way, they are not.
d. This comment builds off the previous one, asking "Is it that certain components of democracy work and democracy as a whole doesn't?" The paper explains that democracy does not work as a check on abuse of power within government. Democracy works as a system for choosing who holds the power of government, but not as a mechanism for limiting the abuse of that power.
I appreciate the references to the literature on veto players, and have integrated that idea, along with the references, into the revised paper. All of the comments were helpful and thought-provoking.
Reviewer 2 Report
A very interesting paper. Very provocative yet balanced. Informative and well done.
Author Response
This reviewer has no suggestions for revision, and says the paper is "Informative and well done." I hope the revision is even more informative.
Reviewer 3 Report
This is an extremely well-written article that highlights the importance of checks and balances over democracy. Here are some extremely minor suggestions for improvement.
It seems to me that the Republican Revolution in 1994 is a great piece of evidence in favor of democracy not working. Even though the House Bank scandal cause the Republican takeover, not much changed with respect to the enforcement of constitutional rules.
Ron Paul seems to me to be the starkest example of disagreements over constitutional interpretation. While it seems to me that he is correct, the other members of Congress still vote (often with near unanimity) to do things that the national government is not specifically authorized to do.
I would have liked to see the author play more of a devil's advocate against checks and balances. Checks and balances work in a framework of rules and often work best when the pie (or power) is fixed. As long as the government is not at Leviathan levels of taxation, however, the branches of government can just agree to raise taxes so that everyone can get the pork they want. It seems to me that the author's argument might be much weaker because checks and balances of representative democracy ultimately have the same problems that democracy has in enforcing constitutional limitations on government - namely that rationally ignorant voters have a difficult time holding the elites accountable for violations.
Zywicki (1997) is a good overview of the public choice interpretation of the 17th Amendment although Tarabar and Hall (2015) don't find empirical support.
Zywicki, Todd J. "Beyond the shell and husk of history: the history of the seventeenth amendment and its implications for current reform proposals." Clev. St. L. Rev. 45 (1997): 165.
Tarabar, Danko, and Joshua C. Hall. "The Seventeenth Amendment, Senate ideology and the growth of government." Applied Economics Letters 22, no. 8 (2015): 637-640.
Author Response
The reviewer makes the comment that different branches of government could just cooperate with each other to provide benefits to each other. This is, indeed, a potential problem, and the way to address it is to have different branches of government have competing interests and a clear division of power, so that each will jealously guard its interests and power from the others. This was done in the original Articles of Confederation, in which the federal government had to requisition the states for funds, and in which the Continental Congress was composed of members appointed by the state legislature. Appropriating funds to the federal government meant taking funds from the states, whose interests were represented by the members of Congress. Similarly, prior to the 17th Amendment in 1913, Senators were chosen by state legislatures, so that any legislation passed by Congress had to be approved by the representatives of the people (in the House of Representatives) and the representatives of the state governments (in the Senate). I'm not saying these systems worked perfectly, but they are examples of how power is not only divided, but also how different holders of power have competing interests.
I appreciate the references, which have been integrated into the paper.